


# Measuring the seismic risk along the Nazca-Southamerican subduction front: Shannon entropy and mutability

Eugenio E. Vogel[1,2], Felipe G. Brevis[1], Denisse Pastén[3,4], Víctor Muñoz[3],
Rodrigo A. Miranda[5,6], Abraham C.-L. Chian[7,8]

[1]Departamento de Física, Universidad de La Frontera, Casilla 54-D, Temuco, Chile
[2]Center for the Development of Nanoscience and Nanotechnology (CEDENNA), 9170124 Santiago, Chile
[3]Departamento de Física, Facultad de Ciencias, Universidad de Chile, Santiago, Chile
[4]Advanced Mining Technology Center (AMTC), Santiago, Chile
[5] UnB-Gama Campus, University of Brasilia, Brasilia DF 70910-900, Brazil.
[6] Plasma Physics Laboratory, Institute of Physics, University of Brasilia, Brasilia DF 70910-900, Brazil.
[7] University of Adelaide, School of Mathematical Sciences, Adelaide, SA 5005, Australia.
[8] National Institute for Space Research (INPE), Sao Jose dos Campos-SP 12227-010, Brazil

*Correspondence to* :eugenio.vogel@ufrontera.cl

**Abstract.** Four geographical zones are defined along the trench that is formed due to the subduction of the Nazca Plate underneath the South American plate; they are denoted A, B, C and D from North to South; zones A, B and D have had a major earthquake after 2010 (8.0), while zone C has not, thus offering a contrast for comparison. For each zone a sequence of intervals between consecutive seisms with magnitudes $\geq 3.0$ is formed and then characterized by Shannon entropy and mutability. These methods show correlation after a major earthquake in what is known as the aftershock regime but they show independence otherwise. Exponential adjustments for these parameters reveal that mutability offers a wider range for the parameters characterizing the recovery to the values of the parameters defining the background activity for each zone before a large earthquake. It is found that the background activity is particularly high for zone A, still recovering for Zone B, reaching values similar to those of Zone A in the case of Zone C (without recent major earthquake) and oscillating around moderate values for Zone D. It is discussed how this can be an indication for more risk of an important future seism in the cases of Zones A and C. The similarities and differences between Shannon entropy and mutability are discussed and explained.

## I. INTRODUCTION

A recent advance on information theory techniques, with the introduction of the concept of mutability (Vogel et al., 2017), opens new ways of looking at the tectonic dynamics in subduction zones. The main goals of the present paper are five-fold: 1) To establish the similarities and differences between mutability and the well-known Shannon entropy to deal with seismic data distributions; 2) To find out which of the aforementioned parameters gives an advantageous description of the subduction dynamics in order to discern different behaviors along the subduction trench; 3) To apply this description to characterize the recovery regime after a major earthquake; 4) To use this approach to establish background activity levels prior to major earthquakes; 5) To apply all of the above to different geographical zones looking for possible indications for regions with indicators pointing for possible future major earthquakes.

Several statistical and numeric techniques have been proposed to analyze seismic events. For a recent review we refer the interested reader to the paper by de Arcangelis et al. and references therein (de Arcangelis et al., 2016). We shall concentrate here in the use of Shannon entropy and mutability which are introduced and discussed in the next paragraphs; they will be applied to the intervals between consecutive seisms in each region.

Data may come from a variety of techniques used to record variations in some earth parameters like infrared spectrum recorded by satellites (Zhang et al., 2019), earth surface displacements measured by Global Positioning System (GPS) (Klein et al., 2018), variations of the earth magnetic field (Cordaro et al., 2018; Venegas-Aravena et al., 2019), among others. In the present work we make use of the seismic sequence itself analyzing the time intervals between filtered consecutive seisms.

Shannon entropy is a useful quantifier for assessing the information content of a complex system (Shannon, 1948). It has been applied to study a variety of nonlinear dynamical phenomena such as magnetic systems, the rayleigh-Bernard convection, 3D MHD model of plasmas, turbulence or seismic time series, among others (Crisanti et al., 1994; Xi et al., 1995; Cakmur et al., 1997; Chian et al., 2010; Miranda et al., 2015; Manshour et al. 2009).

Analysis of the statistical mechanics of earthquakes can provide a physical rationale to the complex properties of



seismic data frequently observed (Vallianatos et al., 2016). A number of studies have shown that the complexity in the content information of earthquakes can be elucidated by Shannon entropy. Telesca et al. (Telesca et al., 2004) applied Shannon entropy to study the 1983-2003 seismicity of Central Italy by comparing the full and the aftershock-depleted catalogues, and found a clear anomalous behaviour in stronger events, which is more evident in the full catalogue than in the aftershock-depleted one. De Santis et al. (De Santis et al., 2011) used Shannon entropy to interpret the physical meaning of the parameter b of the Gutenberg-Richter law that provides a cumulative frequency-magnitude relation for the statistics of the earthquake occurrence. Telesca et al. (Telesca et al., 2012) studied the interevent-time and interevent-distance series of seismic events in Egypt from 2004 to 2010, by varying the depth and the magnitude thresholds.

Telesca et al. (Telesca et al., 2013) combined the measures of the Shannon entropy power and the Fisher information measure to distinguish tsunamigenic and non-tsunamigenic earthquakes in a sample of major earthquakes. Telesca et al. (Telesca et al., 2014) applied the Fisher-Shannon method to confirm the correlation between the properties of the geoelectrical signals and crust deformation in three sites in Taiwan. Nicolis et al. (Nicolis et al., 2015) adopted a combined Shannon entropy and wavelet-based approach to measure the spatial heterogeneity and complexity of spatial point patterns for a catalog of earthquake events in Chile.

Bressan et al. (Bressan et al., 2017) used Shannon entropy and fractal dimension to analyze seismic time series before and after eight moderate earthquakes in Northern Italy and Western Slovenia.

On the other hand, the method based on information theory (Luenberg, 2006; Cover et al., 2006; Roederer 2005) was introduced a decade ago when it was successfully used to detect phase transitions in magnetism (Vogel et al., 2009; Vogel el al., 2012; Cortez et al., 2014). Then a new data compressor was designed to recognize compatible data, namely, data based on specific properties of the system. This method required comparing strings of fixed length and starting always at the same position within the digits defining the stored record. For this reason it was named "word length zipper" (wlzip for short) (Vogel et al., 2012). The successful application of wlzip to the 3D Edwards-Anderson model came immediately afterwards, where one highlight was the confirmation of a reentrant transition that is elusive for some of the other methods (Cortez et al., 2014). Another successful application to critical phenomena was for the disorder to nematic transition that occurs for the depositions of rods of length $k$ (in lattice units) on square lattices: for $k \geq 7$ one specific direction for depositions dominates over when deposition concentration overcomes a critical minimum value (Vogel et al., 2017).

But wlzip proved to be useful not only for the case of phase transitions. It has been used in less drastic data evolution revealing different regimes or behaviors for a variety of systems. The first of such applications were in econophysics dealing with stock markets (Vogel et al., 2014) and pension systems (Vogel et al., 2015). The alteration of the blood pressure parameters was also investigated using wlzip (Contreras et al., 2016). At a completely different time scale the time series involved in wind energy production in Germany was investigated by wlzip yielded recognition of favorable periods for wind energy (Vogel et al., 2018)

The first application of wlzip to seismology came recently using data from a Chilean catalogue finding that wlzip results clearly increase several months prior to large earthquakes (Vogel et al., 2017). The main point in that paper was to establish the method without attempting further analyses or comparison with other methods or to compare possible seismic risk among regions.

In the present paper we make a new analysis comparing results from mutability and Shannon entropy applied to data of seismic data along the subduction zone of the Chilean coast, considering a total number of 22697 seismic events recorded between 2011 and 2017. The complete tectonic context shows an active and complex seismic region for all the coast, driven by the convergence of the Nazca plate and the South American plate, at a rate of 68 mm yr$^{-1}$ (Altamimi et al., 2007) approximately. In the last 100 years, many large earthquakes have been localized in the shock between these two plates, such as Valparaíso 1906 ($M_w$=8.2), Valdivia 1960 ($M_w$=9.6), Cobquecura 2010 ($M_w$ = 8.8), Iquique 2014 ($M_w$=8.2), and Illapel 2015 ($M_w$ =8.4). So, this zone is an attractive source for studying seismic activity associated to large earthquakes. But although the dynamics along the Chilean coast may be dominated by the interaction between these two plates, various works have pointed out variations along the coast which may yield information about the details of that interaction. For instance, the coupling between these two plates has been studied by Metois et al. (Metois et al., 2012; Metois et al. 2013) in the last years, concluding that the subduction area has alternated zones of high an low coupling (Metois et al., 2012; Metois et al., 2013). This suggests that it is interesting to apply novel nonlinear techniques to study such variability. Here, we propose new ways to characterize some of the various dynamics that may be present along the subduction zone in this trench. In order to do that, we will consider four regions along the coast of Chile characterizing them mainly by their latitudes.

The paper is organized in the following way. Next Section is about methodology dealing with the data and parameters to be measured. Section 3 presents the results discussing them and comparing the alternative methods. Last Section is devoted to conclusions.




## II. METHODOLOGY

### A. Data organization

Earthquakes originated in the subduction zone of the Nazca plate underneath the South American plate have been recorded, interpreted and stored in several data seismic data banks. In the present study we shall use the data collected by the Chilean National Seismic Centre (Centro Sismológico Nacional: CNS) (Web, 2019), which are very accurate regarding the location of the epicenters. In particular, we have used a seismic data set collected from March 2005 until March 2017, containing 22 697 events, distributed along the coast of Chile, from Arica in the far north up to Temuco in the south of Chile. These data are freely available through CNS (www.sismologia.cl).

In order to analyze the spatial evolution of the mutability and Shannon entropy along this part of the subduction zone, we have focused our attention on four regions defined below. For each region we have corroborated that the Gutenberg-Richter law holds, finding a common completeness magnitude of $M_w = 3.0$. Thus, all the following analysis will be made using only the seismic events with magnitudes of at least $M_w = 3.0$. We have considered seismic data sequences for four specific geographical zones: three of them include one earthquake over 8.0 occurring after 2010, and we have added for comparison an area with no such large earthquake during several recent years.

Starting from the North, the zones are the following: A) around the earthquake near Iquique (2014; $M_w$ =8.2) including 6891 events; B) around the earthquake near Illapel (2015; $M_w$ =8.4) including 6626 events, C) a quieter geographical region (calm zone) at the center of Chile (where the greatest seismic event is $M_w$ =6.5), including 2824 events; and D) around the earthquake in Cobquecura (2010, $M_w$ =8.8) including 6356 events.The observation time is from January 1, 2011 to March 23, 2017 for zones A, B and C, while it is from January 1, 2009 to March 23, 2017 for zone D (no special reason for this last date). We extended the analysis in the case of zone D to include the regime previous to the big earthquake of 2010. Since the analysis is either relative to the size of the sample or dynamical along the series this difference should not affect the discussion below.

All zones have a similar geographical extension with some singularities that we explain here. Regions A,B, and D have latitudes centered at the epicenter of the largest earthquake of each zone; the span in longitude is the same for these zones. Zone A misses the 4.0° spans in latitude of Zones B and D, since the Chilean catalogue ends at −17.926° which is the northern limit for this zone (for homogeneity of the data we do not mix catalogues). On the other hand, zone C was chosen to include a populated area of the country but with no earthquake over 8.0 and showing less important activity than previous ones. Details are given in Table I, and are illustrated in Fig. 1. As it can be seen in this map Zone C overlaps with both B and D: to avoid getting close to the epicenter of the main earthquake in Zone D, Zone C was shortened in its South extension. So the data catalogues have been filtered by latitude, longitude and magnitude. At this point we do not filter by depth which should not greatly influence the comparison among zones since it is a common criteria for all of them.

|  | Latitudes | | Longitudes | | Main Earthquake | | | |
|------|------------|------------|------------|------------|------|------|---|----|
| Zone | N | S | W | E | Magn | Y | M | D |
| A | −17.926° | −21.572° | −75.00° | −68.00° | 8.2 | 2014 | 4 | 1 |
| B | −29.637° | −33.637° | −75.00° | −68.00° | 8.4 | 2015 | 9 | 16 |
| C | −32.700° | −35.500° | −74.00° | −69.00° | (6.5) | 2012 | 4 | 17 |
| D | −34.290° | −38.290° | −75.00° | −68.00° | 8.8 | 2010 | 2 | 27 |

TABLE I. Geographical definition of the 4 zones considered in this study. The strongest seismic event in each zone is identified at the end. Zone C lacks a very strong seism during recent years which is indicated by the use of parenthesis for the strongest seism here. The geographical coordinates and time windows are explained and defined in the text.

For all seismic events characterized above we calculate the interval in minutes (rounding off seconds) between consecutive events. Then a vector file is produced storing the consecutive intervals between theses seisms within each zone. These are the files to be analyzed by Shannon entropy and mutability.

Let us consider histograms for interval distributions for each zone with consecutive bins of 60 minutes each. Percentage of abundance $G_{k,Z}$ of intervals are obtained for the $k$-th bin for zone $Z$: A, B, C, or D. Figure 2 shows the histograms corresponding to the distribution functions $G_{Z,k}$. It can be immediately seen that shorter intervals have been more frequent in zones D and B, while they are less frequent in the $C$ zone. Zone A presents and intermediate presence of small intervals. This different frequency for small seisms finds an explanation in the presence of large earthquakes in $B$ and $D$ followed by large aftershock periods, while in zone $A$ the aftershock period (and the number of short intervals) was very short as we will see in detail below; Zone $C$ does not include any aftershock period so short intervals are less frequent here.

We can increase the precision of the data treatment below by the use of a data base providing more positions for the

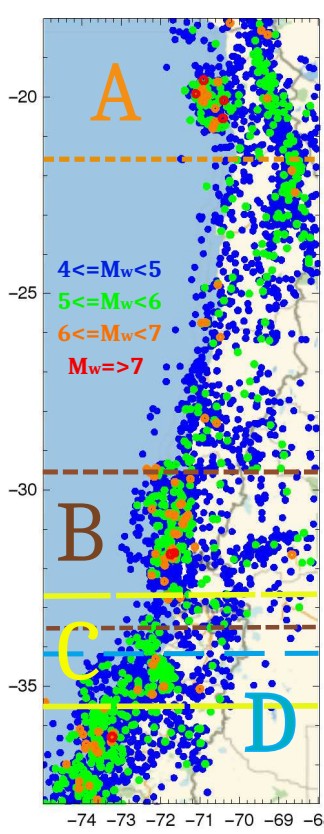

FIG. 1. Map showing the seismic events with magnitudes greater than $M_w$ 4.0 and the division in four geographical zones A, B, C, and D defined in Table I. The seismic events are shown by circles using the following color code according to magnitude: between 4.0 and 4.9 in blue, between 5.0 and 5.9 in green, between 6.0 and 6.9 in orange, and for magnitude equal or greater than $M_w$ 7.0 in red.

numeric recognition (Vogel el al., 2017). This was achieved by choosing a numerical basis providing more positions to be matched. So the data files used both for Shannon entropy and for mutability used digits corresponding to a quaternary numerical basis.

## B.   Shannon entropy

Let $\Delta_i$, $i = 1, ..., N$ be the full sequence of time intervals between consecutive seisms in any of the already defined zones. The time that the $i$-th event occurred can be obtained by $t_i = t_0 + \sum_{j=1}^{i} \Delta_j$, where $t_0$ is the start time of the dataset. The Shannon entropy for $\Delta_i$ within a sliding window of size $\nu$ events can be calculated as follows

$$H(t_i, \nu) = -\sum_{j=i}^{i+\nu} p_j \ln(p_j) \tag{1}$$

where $p_j$ is the probability distribution function of the time intervals within the time window, which can be determined by constructing a normalized histogram

$$p_j = g_j/\nu \tag{2}$$

where $g_j$ is the number of times $\Delta_j$ occurs within the sliding window. For the applications below we use $\nu = 24$.





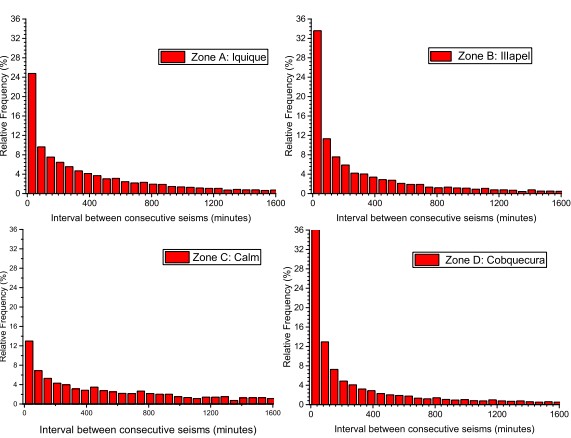

FIG. 2. Distribution functions $G_{Z,i}$ ($Z = \{A, B, C, D\}$) for intervals between two consecutive seismic events.

### C. Data recognizer

We use here the same dynamical data window of $\nu$ instants used for the calculation of Shannon entropy. The relative dynamic information content of a time series of seismic events within a sliding window $\Delta_j$, $j = [i, i + \nu]$, of $w(t_i, \nu)$ bytes is known as mutability, which is defined as

$$\mu(t_i, \nu) = \frac{w^*(t_i, \nu)}{w(t_i, \nu)}, \tag{3}$$

where $w^*$ is the size in bytes of the compressed dataset associated to the time intervals $\Delta_j$ within the time window.

Again we also use $\nu = 24$ for all mutability calculations below. The typical value of $w(t_i, \nu)$ for the files measured here is 144 bytes, while the values of $w^*(t_i, \nu)$ vary roughly between 200 to 400 bytes thus leading to variations in mutability. Two comments are in order: First, wlzip uses compressor algorithms to recognize information but this does not mean that $w^*(t_i, \nu)$ should be less than $w(t_i, \nu)$; Second, the value of wlzip depends both on the interval distribution but also on the time sequence of the intervals while Shannon entropy depends only on the distribution. Thus, the sooner a value in the sequence is repeated, the lower the value of $\mu(t_i, \nu)$ is (vogel et al., 2012; Cortez et al., 2014). This fact marks a difference between these two parameters as we will see below.

### III. RESULTS

Figs. ??, 4, 5 and 6 present the Shannon entropy (top) and mutability (bottom) for data corresponding to geographical areas A, B, C and D respectively according to Table I and Fig. 1. The numeric recognition was done for the data files (intervals in minutes between successive seisms) in quaternary basis both for Shannon entropy and mutability. All registers have the same number of digits filling with zeroes all empty positions previous to first significant digit, The matching to recognize the same data register started at position 4 and was done for three digits including the fourth position (Vogel et al., 2017). All zones were treated with the same precision.

In the upper panel the abscissa "Time" corresponds to real time $t_i$ (as defined in Section II. B) beginning on January 1, 2011 for zones A, B and C, or on January 1, 2009 for zone D. In the lower panel the abscissa labelled "Events" corresponds now to the succession of filtered seisms identified by the same label $i$ used to define $t_i$. The ordinates are the same in both panels.

In the upper panel the aftershock behavior is concealed by the large activity in the short time after a large quake, while in the lower panel it is easier to see the aftershock sequence although the large quiet periods look now more compressed. Earthquakes over a certain magnitude (as given in the inset for each zone) are marked by a star. The





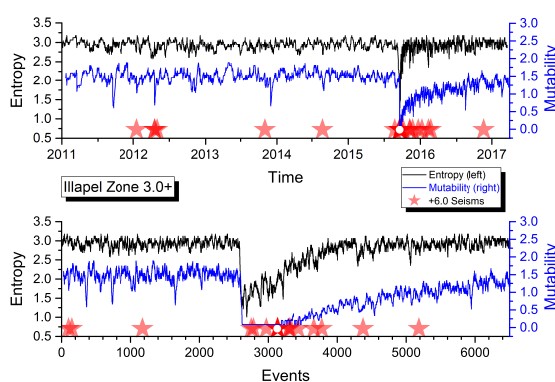

FIG. 4. Shannon entropy and mutability as functions of the sequence of events for the seismic activity of the B zone. The open star marks the position of the earthquake identified in Table I.

As it can be observed, both $H$ and $\mu$ present a similar behavior for the data in the four areas. Immediately after a large shock both indicators sharply decrease due to the short intervals between consecutive aftershocks thereafter.

The average activity level is relatively constant before a major earthquake and later on after the aftershocks have disappeared. However, such activity level is not the same for all the areas which is an indication of different response to similar phenomena which deserves particular attention and it will be further investigated below.

To better appreciate the correlation between $H$ and $\mu$ we study the out-of-phase correlations defined as

$$C_H(n) = \frac{1}{2m+1} \sum_{\ell=-m}^{m} H(n)\mu(n+\ell),\tag{4}$$

$$C_\mu(n) = \frac{1}{2m+1} \sum_{\ell=-m}^{m} \mu(n)H(n+\ell),\tag{5}$$

where $\ell$ is the phase difference measured in terms of number of events separating the measurement of one parameter with respect to the other and $m = 50$ is the maximum phase difference in either sense considered here.

The out-of-phase correlation between Shannon entropy and mutability is presented in Fig. 7: it was found that in general full correlation is lost after about 20 events. A general prevalence is observed in the form of a tendency towards a constant behavior far from the maximum: a value around 0.75 in the wings of zone B (top panel) and towards 0.15 for the zone C (middle panel). Similar figures were analyzed for zones A and D with prevalence values near 0.75 and 0.57 respectively. To test if these prevalence correlations are due to the aftershock regimes a reevaluation of the out-of-phase correlation was done for the D zone restricted to results of Shannon entropy and mutability obtained after January 1, 2013, thus diminishing the effect of the aftershock regime; these results are also shown in Fig. 7 (lower panel). So the main correlation between Shannon entropy and mutability is obtained during the aftershock period. On the other hand the out-of-phase correlation tend to be completely lost during periods without the influence of this regime. This is a first indication for partial independence between Shannon entropy and mutability.

The recovery of the activity level after a major earthquake is faster for the Shannon entropy than for the mutability. Namely, the slope in the recovery for $\mu$ is better defined after a large quake. It is interesting to notice from figures ?? through 6 that zone A recovered its foreshock activity level sooner than any of the other zones. This observation will be put in a quantitative way concentrating on the recovery dynamics in real time to compare the behavior of the different zones.

Figure. 8 presents the mutability results for region D starting at the point of minimum mutability occurring immediately after the major earthquake on February 27, 2010. The dotted (red) curve corresponds to an exponential fit to be discussed next.

For zones A, B, and D, we assume an exponential adjustment for the mutability function after the largest earthquake. A possible such function is:

$$\mu_{eZ}(t) = a_Z + b_Z \exp(-(t - t_Z)/\tau_Z),\tag{6}$$

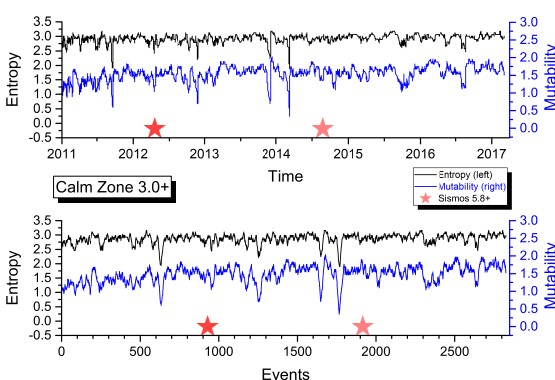

FIG. 5. Shannon entropy and mutability as functions of the sequence of events for the seismic activity of the C zone. The open star marks the position of the earthquake identified in Table I.

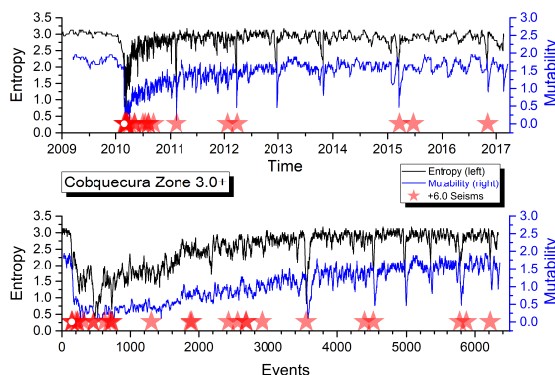

FIG. 6. Shannon entropy and mutability as functions of the sequence of events for the seismic activity of the D zone. The open star marks the position of the earthquake identified in Table I.

where $a_Z$ measures the "asymptotic" activity of zone $Z$ (reached after the aftershocks regime), $t_Z$ corresponds to the time of minimum mutability after the largest earthquake (Table I) and serves as initial time for this recovery analysis; $\tau_Z$ is the characteristic time for activity recovery in zone $Z$. $b_Z$ is just a shape adjustment parameter without a direct meaning for this analysis.

For zone D (Fig. 8), the best least square fit for the mutability is obtained for $a_D = 1.502(2)$ and $\tau_D = 0.62212$ years (y). The results of this treatment for all the zones with major earthquakes are summarized in Table II.

| Zone $Z$ | $a_Z$ | $b_Z$ | $t_Z$ (yr) | $\tau_Z$ (y) |
|---|---|---|---|---|
| A | 1.754 (0.002) | −1.64691 | 2014.24829 | 0.0134(3) |
| B | 1.208 (0.004) | −1.09124 | 2015.70784 | 0.2092(33) |
| D | 1.502 (0.005) | −1.37833 | 2010.07093 | 0.6221(110) |

TABLE II. Best fit parameters for the mutability of zones A, B, and D, after the main earthquake, using the exponential trial function given by Eq. (6).

A similar analysis was made for the Shannon entropy results using the same exponential fit and the corresponding parameters are given in Table III.

Figures similar to Fig. 8 were made for the mutability of zones A and B using the best fit parameters listed





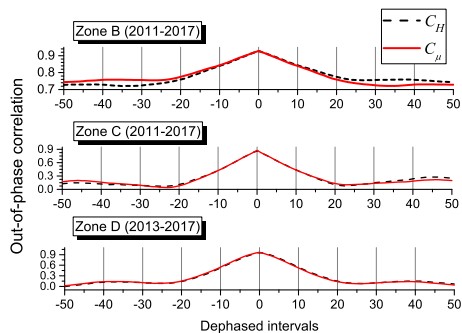

FIG. 7. Out-of-phase correlations. Upper panel: B zone data including aftershock regime (similar ones are obtained for zones A and D with aftershock regimes). Middle panel: C zone data that does not present aftershock regime. Lower panel: Truncated D zone data to exclude the aftershock regime.

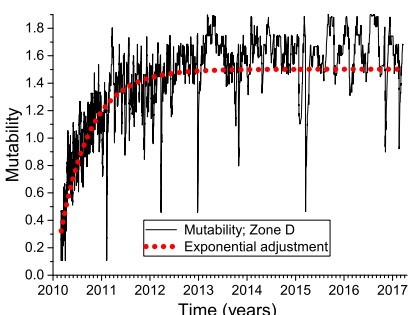

FIG. 8. Exponential fit for Cobquecura data set after February 27, 2010. This data set starts at the point of minimum mutability after the big earthquake of magnitude $M_w = 8.8$.

in Table II. The same analysis was also done for the results obtained by Shannon entropy and the corresponding parameters are given in Table III. The figures backing such fittings are not included here since they are very similar to Fig. 8 and the procedure is the same to the one already established in the presentation of this figure.

Let us now discuss the results given in Tables II and III. The first striking difference between Shannon entropy and mutability is on the value for the background parameter $a_Z$. In the case of the adjustment for Shannon entropy this parameter does not discriminate significantly among zones with values close to 2.9 for all of them; the same parameter in the case of the mutability data spans a range [1.208,1.754], thus indicating differences in the dynamics of these three regions. In particular, mutability indicates that in Zone B there are more seismic events at regular intervals than in the other zones. Given the underlying plate subduction mechanism, this could mean that plates are sliding more regularly or even fluently in Zone B, whereas the relative motion of the Nazca plate under the South-American plate is more difficult in Zone A, thus leading to more disperse set of intervals between consecutive seisms.

After a large earthquake the zones tend to recover their characteristic activity level $a_Z$, but this is done rather abruptly for Shannon entropy while it is more gradual for mutability. This is measured by the recovery time $\tau_Z$ in Tables II and III. In the case of the Shannon entropy for the Zone A the recovery is very fast, namely 0.00947 years $\approx$ 3.5 days. In the case of Zones B and D the recovery times for the Shannon entropy are of 9 days and 45 days, respectively. However, when the analysis is done using the recovery time for mutability (Table II) the recovery times are 5 days, 2.5 months and 7.5 months for the zones A, B, and D, respectively.

Tables II and III also show that recovery times $\tau_Z$ are different, shorter for Shannon entropy and longer for mutability, but the tendencies are the same. So eventually both methods can be used to characterize this aspect of the aftershock regime. In terms of the human perception experienced after any large earthquake it seems that $\tau_Z$ values obtained for the mutability results are more representative of the aftershock times experienced in each zone.




| Zone $Z$ | $a_Z$ | $b_Z$ | $t_Z$ (yr) | $\tau_Z$ (yr) |
|---|---|---|---|---|
| A | 2.924(2) | −3.69218 | 2014.24516 | 0.0095(3) |
| B | 2.908(3) | −2.30957 | 2015.69997 | 0.0246(4) |
| D | 2.815(4) | −2.29226 | 2010.13133 | 0.1255(25) |

TABLE III. Best fit parameters for Shannon entropy of zones A, B, and D, after the main earthquake, using the exponential trial function Eq. (6).

Thus, for instance, seisms of magnitude around 4.0 were frequent in zone D during several months after February 10, 2010, but this was not the case for zone A where people lost perception of the aftershock regime after a week or so of the last earthquake in this area.

The main difference between Shannon entropy and mutability is that the former analyzes the distribution of registers in a sequence regardless of the order in which these entries were obtained, while the latter gives a lower result for sequences including frequently repeated registers (Cortez et al., 2014). Shannon entropy considers the visit to a state without considering the order in which these visits take place, so it pays exclusive attention to the probability of visiting a state at some instance during the observation time. Mutability considers also the trajectory in which these visits take place, giving lower results when the system stays long periods in the same state or states directly connected to this state; on the contrary during agitated periods (chaotic regimes would the at the apex here) mutability gives higher results. In other words, a given sequence has just one result for Shannon entropy but the permutations of the order of the registers lead to different results for mutability; in the present case the mutability results reported here corresponds to the natural sequence of the recorded seisms.

We now focus on the analysis of the background activity obtained for the 4 zones described in this work, taking semestral averages of the values of mutability in Figs. ??–6, in order to study trends in time scales longer than the one of previous figures. We have chosen a semester as the time for averages so we have a few hundreds registers in each partial sequence minimizing error, but still we have some 13 points in the overall period to appreciate tendencies and differences. In doing so, we also evaluate semestral averages of intervals between consecutive seisms, which show similar trends to the mutability results for the same period.

The semestral analysis for zones A, B, C and D is shown in Figs. 9, 10, 11 and 12, respectively; they are all presented under the same scale to allow a direct comparison. The mutability values run on the upper part (black) while the intervals tend to occupy the lower part (blue) of the plot. The first comment here is evident: these 4 regions present different seismic behaviors so we have to discuss them separately. The only common feature is that an earthquake with magnitude over 8.0 produces an absolute minimum for both variables during the semester containing this seism and its aftershock sequence.

For didactical reasons we shall perform this discussion beginning with zone D, where the long recovery period already appreciated in Fig. 6 and in Table II is more enhanced. It is interesting to observe that the average semestral mutability presents recent relaxations like in the first semester of 2015 and the first semester of 2017. Generally speaking these results do not approach yet the values near 1.8 for the average semestral mutability in the foreshock period preceding the large earthquake of 2010. Interval semestral averages tend to follow the variations of mutability but some differences are noticed. The present average interval of about 2000 minutes (about 33 hours) is far from the almost 6000 minute interval before the large earthquake.

Fig. 11 is completely different to the others. There is no major earthquake included here but it is obvious that there was one prior to 2011 from which this activity is slowly recovering. The general tendency is to slowly increase the mutability values to levels similar to those constantly presented by zone A and those presented by region D previous to the large earthquake. Interval averages also increase reaching just under 2000 minutes. If this is an announcement for a future major earthquake in Zone C or nearby is still too early to tell but this zone should be monitored closely.

Fig. 10 shows the foreshock mutability averages for Zone B which present a nearly flat behavior around 1.6 before the major earthquake of 2015. Then, after the aftershock regime the average semestral mutability begins to recover, faster than in zone D, but still not reaching the level shown here previous to the large earthquake. The observation is similar for the interval semestral average whose value is still small compared to the activity before 2016.

Fig. 9 shows the almost constant results (near 1.8) for the average semestral mutability of zone A, with just one semester reaching a moderate low value (1.4 with large error bar).The semestral average for intervals between seisms is also rather flat around 10 hours. The only exception is the first semester of 2014 in coincidence with the large earthquake there.

Error bars deserve a separate discussion. In the case of mutability they are rather small for the A Zone, meaning that the intervals are rather similar along the data sequence. This is reinforced by the average interval error bars which are the smallest among the four zones (spanning only about 1200 minutes) telling that intervals are not so different among themselves. The largest error bars both for mutability and intervals are to be found in Zone D; moreover they are irregular in recent years. Error bars increased for the average in zone D during 2009 just prior to the huge quake





of 2010. However, for this same zone the corresponding error bars for the average semestral mutability are among the smallest to be found prior to this large earthquake. Once again it is difficult to say something about the present status of Zone B since it is clearly under recovery. However, the Calm Zone C is clearly showing a tendency: error bars for mutability averages are shrinking, while error bars for intervals are growing spanning about 60 hours. These two symptoms were present in zones A, B and D previous to their large respective earthquakes. In the case of Zone A the error bars for the average intervals are not so large, but here is where we find the highest values for mutability and the smallest error bars for this variable.

If we look for common features just before a large earthquake they are: relatively high mutability values ("high" needs to be defined for each zone) and very small error bars associated with semestral mutability averages. The particular values of these indicators for Zone A could be interpreted as an irregular subduction here, with no short-time accommodations or lack of fluency, leading to seismic risk of some sort, although it is not possible to specify any possible time for a large seism in the future. From this point of view, the earthquake of 2014 near Iquique was just a small accommodation of the plates but the subduction process could be somewhat stuck to the similar levels presented before the large quake.

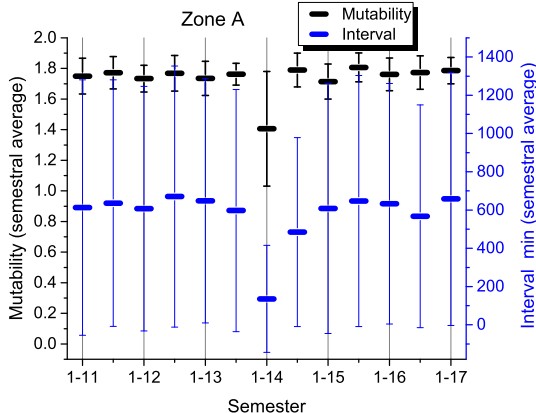

FIG. 9. Semestral average of mutability values (upper symbols; black) and intervals in minutes between consecutive seisms (lower symbols; blue) for Zone A: Iquique. Odd semesters are labeled on the abscissa axis (1-13: first semester of 2013) while even semesters are only marked.

## IV. CONCLUSIONS

Seismic activity is different for the four zones defined here along the Nazca-South American subduction trench (Figs. 1-2, Table I). Nevertheless, some general behaviors are common to the seismicity of the tectonic activity present in this region. Both Shannon entropy and mutability show a sudden decrease after an earthquake of magnitude around or over 7.0 (Figs. ??-6). Additionally, Shannon entropy and mutability reach "high" values before a major earthquake; the scale to define "high" needs to be tuned for each geographical region and observation time window.

A short time correlation exists between Shannon entropy and mutability during the aftershock regime. However, this correlation is lost far from this regime thus providing independent tests to characterize the seismic activity (Fig. 7).

The aftershock regime is characterized by successions of low and medium intensity seisms at short intervals producing low values of both Shannon entropy and mutability. After some recovery time the intervals tend to go back to the kind of intervals present before the large quake. This recovery behavior can be described by exponential adjustments (Fig. 8) which indicate that the characteristic times are longer for mutability than for Shannon entropy (Tables II-III); eventually this speaks in favor of the former to continue the analysis. Another advantage of mutability is that the parameter reflecting the background activity span larger ranges than the one presented by the adjustment of Shannon entropy (Tables II-III). From these results the mutability recovery time $\tau_Z$ for Zone A lasted a few days, while the same parameters for Zone D lasted several months, which is close to the human perception in these zones.

The differences between Shannon entropy and mutability evidenced after the recovery time are due to the handling of

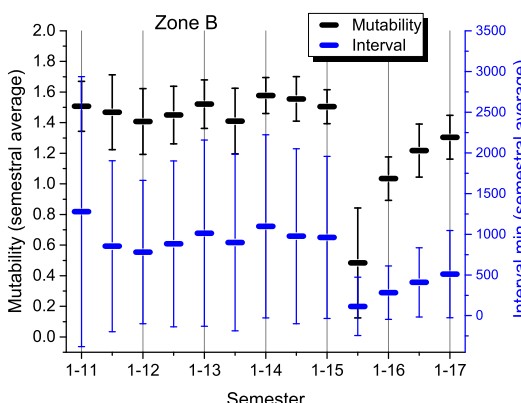

FIG. 10. Semestral average of mutability values (upper symbols; black) and intervals in minutes between consecutive seisms (lower symbols; blue) for Zone B: Illapel. Odd semesters are labeled on the abscissa axis (1-13: first semester of 2013) while even semesters are only marked.

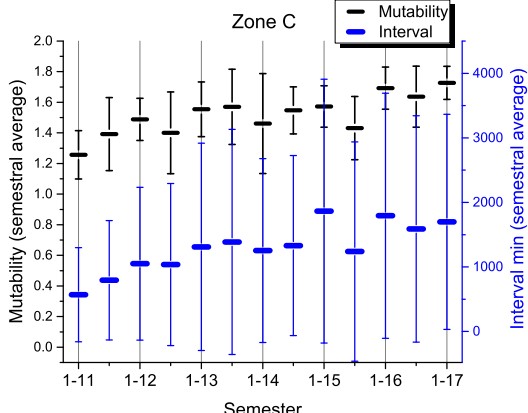

FIG. 11. Semestral average of mutability values (upper symbols; black) and intervals in minutes between consecutive seisms (lower symbols; blue) for Zone C: Calm. Odd semesters are labeled on the abscissa axis (1-13: first semester of 2013) while even semesters are only marked.

a static distribution by the former while the latter considers the order in which the registers entered in the distribution.
From this point of view mutability carries more information than Shannon entropy.
The background activity based on mutability $a_Z$ (Tables II-III) is quite different for each zone (Figs. 9-12). This
means that the subduction process finds different difficulties in each zone. However, some general features describing
the motion of the Nazca plate under the South-American plate should be present along the trench. To investigate
this possibility we considered semestral averages of mutability values.
Semestral averages for mutability recovered soon for Zone A after the 8.2 earthquake, which indicates that the
short intervals after a major earthquake were mostly absent here. Soon, the regime with longer and different intervals
reappeared raising the values of mutability and narrowing the corresponding error bars; this could be interpreted as
a warning for a possible earthquake in this zone sometime in the near future. On the opposite side is Zone D where
the semestral averages still do not recover to the levels prior to the large 8.8 earthquake of 2010; moreover, there have
been instances lowering the semestral averages for mutability with large error bars in recent times evidencing short
intervals indicating activity in a rather continuous way. In the case of Zone B the recovery is still under way so it is





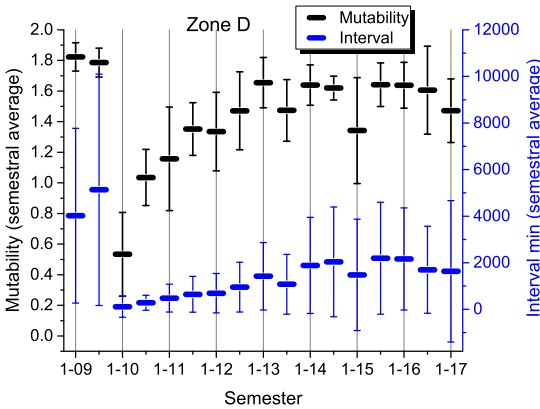

FIG. 12. Semestral average of mutability values (upper symbols; black) and intervals in minutes between consecutive seisms (lower symbols; blue) for Zone D: Cobquecura. Odd semesters are labeled on the abscissa axis (1-13: first semester of 2013) while even semesters are only marked.

too soon to say anything at this time. Generally speaking we can observe that mutability values were high and their error bars were small just before a major earthquake in zones A, B, and D.

Semestral averages for intervals between consecutive seisms and their corresponding error bars are very different among the different regions. Both values decrease during the aftershock regime but no clear trend could be found prior to a large earthquake.

As for the Calm Zone C the mutability semestral averages are clearly increasing reaching 1.8 with narrowing error bars. Although each zone can have different thresholds for triggering of a major event, such value or slightly lower ones have been present just before large earthquakes in the other zones. Eventually Zone C is showing a behavior that should be further studied at the expectation of future large quakes.

Let us close by answering the 5 points raised in the introduction thus summarizing previous discussions and conclusions. 1) Both Shannon entropy and mutability give similar responses to a major earthquake and its immediate aftershock period, however they are independent and non-correlated during the quieter periods. 2) Shannon entropy deals with the distribution as a whole while mutability deals with a sequential distribution; this allows to the latter be more effective in providing larger contrasts if the values of the characteristics parameters. 3) The recovery time and background activity are very well characterized by mutability allowing to discriminate among different zones. 4) The mutability semestral averages reflect the seismic activity of the different zones indicating where the subduction is relatively fluent or where the process could be stuck. 5) A combined analysis points to Zone A as stuck for many years and Zone C slowly decreasing fluency in the subduction process which can be indication for accumulation of energy in this zone.

## ACKNOWLEDGEMENTS

One of us (EEV) is grateful to Fondecyt (Chile) under contracts 1150019 and 1190036, and Center for the Development of Nanoscience and Nanotechnology (CEDENNA) funded by Conicyt (Chile) under contract FB0807 for partial support. DP thanks Advanced Mining Technology Center (AMTC) and the Fondecyt grant 11160452. VM thanks Fondecyt project 1161711.

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
