# Peer review of "Measuring the seismic risk along the Nazca-Southamerican subduction front: Shannon entropy and mutability"

_Natural Hazards and Earth System Sciences, 2019_

## Referee Comment (RC1) · Anonymous Referee #1 · 21 Dec 2019

General Comments: In this manuscript (ms), Vogel et al. apply Shannon entropy and mutability for the estimation of the seismic risk along the Nazca-Southamerican subduction front. Four geographical zones are selected along the trench formed by the subduction of the Nazca Plate under the South American plate. The authors study the sequence of intervals between consecutive earthquakes (EQs) in each of the four geographical zones by using Shannon entropy and mutability. The obtained results are interesting but the presentation does not conform to the existing literature although it uses ideas earlier published by other researchers.

Specific Comments: For example, in both methods, see, e.g., Eqs. (1), (4) and (5),

the number of events (or event windows) is used in the sense it is used in natural time analysis that has appeared almost two decades ago, see, e.g.

[P. Varotsos, N. Sarlis, and E. Skordas, Spatiotemporal complexity aspects on the inter-relation between Seismic Electric Signals and seismicity, Practica of Athens Academy, 76, 294-321, 2001. Available from http://physlab.phys.uoa.gr/org/pdf/p3.pdf]

[P.A. Varotsos, N.V. Sarlis, and E.S. Skordas, Long-range correlations in the electric signals that precede rupture, Phys. Rev. E, 66, 011902 (7), 2002.]

[Varotsos P.A., Sarlis N.V. and Skordas E.S., Natural Time Analysis: The new view of time. Precursory Seismic Electric Signals, Earthquakes and other Complex Time-Series (Springer-Verlag, Berlin Heidelberg) 2011]

but the original works are not mentioned. The importance of natural time in the study of seismicity has been recently stressed by

[Rundle, J. B., D. L. Turcotte, A. Donnellan, L. Grant Ludwig, M. Luginbuhl, and G.Gong (2016), Nowcasting earthquakes, Earth and Space Science, 3, 480–486, doi:10.1002/2016EA000185],

I quote "Event counts as a measure of "time," rather than the clock time, is known as "natural" time [Varotsos et al., 2002, 2005, 2011; Holliday et al., 2006]. We will show that the use of natural time has at least two advantages when applied to earthquake seismicity. . .".

Thus, the authors should accommodate their present findings within the pre-existing literature by inserting in the Introduction section in page 2 a paragraph concerning the findings of natural time analysis related to seismicity. Indicative results can be found in the references mentioned in the above quotation as well as the more recent:

[P.A. Varotsos, N. V. Sarlis, E. S. Skordas, Seiya Uyeda and Masashi Kamogawa, "Natural time analysis of critical phenomena", Proceedings of the National Academy of Sciences of the United States of America, Vol.108 (2011), 11361-11364.]

[N. V. Sarlis, E. S. Skordas, P. A. Varotsos, T. Nagao, M. Kamogawa, H. Tanaka, and S. Uyeda, "Minimum of the order parameter fluctuations of seismicity before major earthquakes in Japan", Proceedings of the National Academy of Sciences of the United States of America, Vol.110 (2013), 13734–13738.]

[N. V. Sarlis, E. S. Skordas, P. A. Varotsos, T. Nagao, M. Kamogawa, and S. Uyeda, "Spatiotemporal variations of seismicity before major earthquakes in the Japanese area and their relation with the epicentral locations", Proceedings of the National Academy of Sciences of the United States of America, Vol.112 (2015), 986–989.]

[N.V. Sarlis, E.S. Skordas, A. Mintzelas, and K.A. Papadopoulou, "Micro-scale, mid-scale, and macro-scale in global seismicity identified by empirical mode decomposition and their multifractal characteristics", Scientific Reports, Vol. 8 (2018), 9206, DOI 10.1038/s41598-018-27567-y]

[N. V. Sarlis, E. S. Skordas, and P. A. Varotsos, "A remarkable change of the entropy of seismicity in natural time under time reversal before the super-giant M9 Tohoku earthquake on 11 March 2011", EPL, Vol. 124 (2018), 29001(7), DOI 10.1209/0295-5075/124/29001.]

[J. B. Rundle, M. Luginbuhl, A. Giguere, D. L. Turcotte, Natural Time, Nowcasting and the Physics of Earthquakes: Estimation of Seismic Risk to Global Megacities, Pure and Applied Geophysics 175 (2018) 647-660. doi:10.1007/s00024-017-1720-x.]

[J. B. Rundle, A. Giguere, D. L. Turcotte, J. P. Crutchfield, A. Donnellan, Global Seismic Nowcasting With Shannon Information Entropy, Earth and Space Science 6 (1) (2019) 191-197. doi:10.1029/2018EA000464.]

Appropriate changes should be also made in line 334 where it is written "while the latter considers the order in which the registers entered in the distribution" but natural time is not mentioned, and in lines 359-360 in the Conclusions. What the authors state here is one of the major applications of natural time analysis and in particular for the

entropy change under time reversal, see, e.g.,

[P. Varotsos, N. Sarlis, E. Skordas, and M. Lazaridou, "Identifying sudden cardiac death risk and specifying its occurrence time by analyzing electrocardiograms in natural time", Applied Physics Letters, Vol. 91 (2007), 064106]

Additionally, on Page 1, lines 43 to 46 the Authors write: "Data may come from a variety of techniques used to record variations in some earth parameters like infrared spectrum recorded by satellites (Zhang et al., 2019), earth surface displacements measured by Global Positioning System (GPS) (Klein et al., 2018), variations of the earth magnetic field (Cordaro et al., 2018; Venegas-Aravena et al., 2019), among others." I cannot understand completely the meaning of the sentence, hence it needs rewording. If it refers to precursory changes before EQs there is an obvious omission of the Seismic Electric Signals that precede EQs, see, e.g.,

[P. Varotsos and K. Alexopoulos, Physical properties of the variations of the electric field of the earth preceding earthquakes, I. Tectonophysics 110, 73-98, 1984.]

[P. Varotsos and K. Alexopoulos, Physical properties of the variations of the electric field of the earth preceding earthquakes, II. Determination of epicenter and magnitude, Tectonophysics 110, 99-125, 1984.]

[P. Varotsos, K. Alexopoulos, K. Nomicos and M. Lazaridou, Earthquake prediction and electric signals, Nature 322, 120, 1986.]

[P. Varotsos and M. Lazaridou, Latest aspects of earthquake Prediction in Greece based on Seismic Electric Signals. I, Tectonophysics 188, 321-347, 1991.]

[P. Varotsos, K. Alexopoulos and M. Lazaridou, Latest aspects of earthquake prediction in Greece based on Seismic Electric Signals II, Tectonophysics 224, 1-37 1993.]

[P. Varotsos, The Physics of Seismic Electric Signals, TerraPub, Tokyo (2005) 338 pages.]

[N.V. Sarlis, P.A. Varotsos, E. S. Skordas, S. Uyeda, J. Zlotnicki, T. Nagao, A. Rybin, M.S. Lazaridou-Varotsos, and K.A. Papadopoulou, "Seismic Electric Signals in seismic prone areas", Earthquake Science, Vol. 31 (2018), 44-51, DOI 10.29382/eqs-2018-0005-5.]

[P.A. Varotsos, N.V. Sarlis and E.S. Skordas," Phenomena preceding major earth-quakes interconnected through a physical model", Annales Geophysicae 37 (2019), 315–324.]

Technical Corrections: I am now turning to other problems with the presentation:

Page 2, lines 145 and 146 two different symbols appear for G_{k,Z}, G_{Z,k} also in Figure Caption 2.

Page 2, line 152: "data base" -> "database"

Page 4, line 162: why \nu was selected 24?

Page 5, line 166: Please give an explicit definition of what is meant by "w(t_i, \nu ) bytes"?

Page 5, line 167: Also provide an explicit definition of wˆ* because I cannot comprehend the term "wˆ* is the size in bytes of the compressed dataset associated to the time intervals \Delta_j within the time window." What is a compressed dataset for a time interval? and how time written when uncompressed?

Page 5, line 176: Figure 3 is missing, also in many lines in the paper e.g. line 213 on page 6.

Page 5, lines 183-184: It should be clarified here that the figure depicting t_i uses natural time.

Page 5, at the end: lines 189-190 are missing.

Page 6, line 217: "Figure." -> "Figure"

Page 7, Table II, first row, fourth column: "(yr)" -> "(y)"

Page 9, line 285: For the readers' convenience please mention the EQ to which you are referring to.

Page 9, line 297: Please explain how the error bars were found.

Page 10, Figure 9: The error bars drawn in the figure reach even negative values of the interoccurrence interval

Page 11, line 334: I cannot understand the term "the registers entered in the distribution", please explain.

In summary, I suggest that the authors make a major revision along the lines suggested above.
* * *

---

## Short Comment (SC1) · 30 Dec 2019

Very interesting application of Shannon entropy to explore earthquake detonations! In some cases the earthquakes are observed close to the temporal proximity of abrupt entropy changes. However, in other cases there seems to be no correlation.

---

## Referee Comment (RC2) · Anonymous Referee #2 · 3 Jan 2020

The manuscript studies the evolution of seismicity in four different regions of the Nazca plate. In my opinion the main result is the very slow recovery time exhibited by the mutability after the largest mainshock, with significant differences in its specific value observed in the three different regions. This result can be very interesting but, unfortunately, I am not able to appreciate it, in the present form.

My main problem is the understanding of the meaning of the mutability. My first suggestion is to give a more detailed definition of $\mu$ with also some practical example of how to measure it. My second suggestion is to study this quantity for a synthetic catalog. More precisely, the main statistical features of aftershock clustering is captured by the ETAS model (see for instance de Arcangelis et al., Phys. Rep. 2016 and references therein). The ETAS model has only one characteristic time scale which is the occurrence rate of independent mainshocks and therefore it is very reasonable to explore if the slow recovery time is also found in synthetic catalogs. In the case of a positive answer its specific value must be related to the mainshock occurrence rate. More interestingly would be the case of a negative answer, which suggests some feature not captured by the ETAS model, like incompleteness (de Arcangelis et al., Journ. Geophys. Res., 2018), magnitude correlations (Lippiello et al., Phys. Rev. Lett. 2007), foreshocks (Lippiello et al., Entropy, 2019) ....

Minor points

- Figure 3 is missing; - In general I suggest the use of a semi-log scale in Fig.s 2,7,8. In particular it would be interesting to check if the relaxation to the background level of $\mu$ (Fig.8) is also compatible with the power law decay reminiscent of the Omori law; - For Fig.4-6 I don't find any particular useful information in plotting the quantities as function of both $t_i$ and $i$. I suggest to chose just one of the two panels and to plot in the same figure the results for the four regions; - The open star is not easily found in Fig.4 and Fig.6, i suggest to use a different symbol. The open star should also be not present in Fig.5. Please, correct the caption; - Is it possible to motivate the choice for $m=50$ in the definition of $C$ and, more importantly, for $\ni=24$ in the definition of entropy and mutability? In particular, I find important to discuss, and eventually show, how results are affected by the precise value of $\ni$. - The interval distribution plotted in Fig.2 is more commonly defined as intertime or inter-event time distribution. I suggest to rescale the horizontal axis in Fig.2 by the average occurrence rate of each region. This should allows to stress deviation from the universal scaling behaviors (see Lippiello et al., Phys. Rev. E, 2012); - I believe that it is difficult to extrapolate the the stationary value of the mutability from the best fit of Eq.6 since it appears that $\mu$ is not yet relaxed to its stationary value. I suggest to also evaluate the average value of $\mu$ before the mainshock occurrence. - I also suggest to consider the connected

correlation function, by subtracting the average value $<H><\mu>$.

---

## Author Comment (AC1) · 30 Jan 2020

Thank you for your valuable comments all of which have been taken into account in the present version of the paper. We explicitly mention natural time now in the text and in the figures; previous literature is quoted. We notice, though, that in our case, temporal information is kept, since vector files analyzed contain the interevent times. So there are similarities with the "natural time" concept, since vectors are indexed by the event number, but there are also subtleties that we believe should be taken into account as well. The paragraph beginning with "Data may come ..." in Page 1 has now been extended to include other related data sources. We have also rephrased

some parts to clarify the presentation. The notation problem Concerning $G_{Z,k}$ has been settled. Other typos you mention also were corrected. Thank you. Yes, you are right concerning the lack of discussion for the time window $\nu$. We have written now a full paragraph (previous to Section IIC), quoting two references, to justify this choice. Additionally, the examples given in the Appendix all used a time window of 24 instants to better appreciate this is an appropriate choice. The already mentioned Appendix comes in support of a more extended presentation on mutability, defining explicitly w and w* as you ask. We did not do that in previous version since in the literature there are already thorough presentations with examples. However, for the benefit of the audience it could be convenient to have examples more related to the present kind of data. The ways the error bars are obtained are now explained including the way the semester averages are calculated, discussing the extreme values.
* * *

---

## Author Comment (AC2) · 30 Jan 2020

Thank you for your comment. The data recognizer has a tuning mechanism that allows to set the threshold response. In the present case we have adjust it to recognize important earthquakes.

---

## Author Comment (AC3) · 30 Jan 2020

Thank you for your valuable suggestions all of which have been pondered upon writing the present version of the paper. Concerning what you call "main problem" in relation to understanding mutability we follow your concern and we have expanded the presentation of mutability to the point of including an Appendix with 4 examples which we hope help to better grasp the concept. We also quote specific aspects of previous literature which can also help the reader to better follow the discussion. Thank you for raising this point. Concerning what you call your "second suggestion" we believe that it would take us beyond the goals intended for this paper. Our main aim is to use one recently introduced method to find differences and possible seismic risk for different zones in the Nazca-Southamerican trench. If we additionally include comparison among methods this would make a different, heterogeneous and much longer paper. So we have not included this suggestion in the present version although we have added a paragraph at the end of the paper, commenting on this issue. We now turn our attention to what you call "Minor points" in the order you wrote them. - Yes, Fig. 3 was missing. Sorry about that. Now is back. - We follow your suggestion and use semilog scale in Figs. 2 and 8. It does not seem to be necessary in Fig. 7. Discussions have been updated accordingly. Present Fig. 2 looks much more informative than previous one. Thank you for that. As for Fig. 8 we have included the semilog plot as an inset where the resemblance with a power law is explicitly mentioned. - We quote the reference to the scaling of interevent intervals although this alternative treatment is out of the main scope of the present paper based on information theory of the natural interevent intervals. - With respect to the double panels in Figs. 3 through 6 we believe some readers can benefit from the double presentation. This is even more so since the other referee points to the need of explaining and stressing the importance of "natural time" (what you call "intertime or inter-event"). So this panel needs to be included anyhow but losing any reference to real time to properly identify the real time scale. Since there is no strong argument to get rid of real time in the figures, we leave them in the double presentation but changing the open star to a different symbol to avoid confusions. Caption of Fig. 5 was also corrected. - The value m=50 is empirical and obtained so a flat behavior of the correlation functions is reached. This is now discussed under the equations defining the correlations. - Several lines have been added to the paragraph immediately under Eq. (2) to justify the choice $\nu=24$. The comparison among different time windows was done elsewhere and one helpful figure is now quoted. Additionally, the examples given in the Appendix all used a time window of 24 instants to better appreciate this is an appropriate choice. - Yes you are right, it takes quite some time to attain any "stationary value" for the mutability, so we can only deal with approximations to that ideal regime. - Your suggestion of evaluating the average value of $\mu$ before the

mainshock is done in Figs. 9, 10 and 12 in the case of semestral averages. It is not clear if any other information could be obtained for a different period before the main earthquakes. - Thank you for pointing to subtract the average values <H> and <mu>. This is now included in present Eqs. (4) and (5). In previous version these equation were in an incomplete form, although the plots were right.

———————————————————